# Imaging of super-fast dynamics and flow instabilities of superconducting vortices

L. Embon[1], Y. Anahory[1,2], Ž.L. Jelić[3,4], E.O. Lachman[1], Y. Myasoedov[1], M.E. Huber[5], G.P. Mikitik[6], A.V. Silhanek[4], M.V. Milošević[3], A. Gurevich[7] & E. Zeldov [1]

Quantized magnetic vortices driven by electric current determine key electromagnetic properties of superconductors. While the dynamic behavior of slow vortices has been thoroughly investigated, the physics of ultrafast vortices under strong currents remains largely unexplored. Here, we use a nanoscale scanning superconducting quantum interference device to image vortices penetrating into a superconducting Pb film at rates of tens of GHz and moving with velocities of up to tens of km/s, which are not only much larger than the speed of sound but also exceed the pair-breaking speed limit of superconducting condensate. These experiments reveal formation of mesoscopic vortex channels which undergo cascades of bifurcations as the current and magnetic field increase. Our numerical simulations predict metamorphosis of fast Abrikosov vortices into mixed Abrikosov-Josephson vortices at even higher velocities. This work offers an insight into the fundamental physics of dynamic vortex states of superconductors at high current densities, crucial for many applications.

[1] Department of Condensed Matter Physics, Weizmann Institute of Science, Rehovot 7610001, Israel. [2] The Racah Institute of Physics, The Hebrew University of Jerusalem, Jerusalem 9190401, Israel. [3] Departement Fysica, Universiteit Antwerpen, Groenenborgerlaan 171, B-2020 Antwerpen, Belgium. [4] Département de Physique, Université de Liège, B-4000 Sart Tilman, Belgium. [5] Departments of Physics and Electrical Engineering, University of Colorado Denver, Denver, Colorado 80217, USA. [6] Verkin Institute for Low Temperature Physics & Engineering, Ukrainian Academy of Sciences, Kharkov 61103, Ukraine. [7] Department of Physics, Old Dominion University, Norfolk, Virginia 23529-0116, USA. L. Embon, Y. Anahory and Ž.L. Jelić contributed equally to this work. Correspondence and requests for materials should be addressed to Y.A. (email: yonathan.anahory@mail.huji.ac.il) or to E.Z. (email: eli.zeldov@weizmann.ac.il)

The dynamics of current-driven vortex matter is of major importance both for the comprehension of the fundamental collective behavior of strongly interacting vortices and for attaining high non-dissipative currents in superconductors for applications. Materials advances in incorporating artificial pinning centers that immobilize vortices, particularly oxide nano-precipitates in cuprates, have resulted in critical current densities $J_c$ as high as 10–20% of the depairing current density $J_d$ at which the superconducting state breaks down[1–3]. At such high current densities $J$, once a vortex gets depinned from a defect, it can move with high velocity $v$ and dissipate much power. Understanding this phenomenon is critical for many applications, such as high-field magnets[4], superconducting digital memory and qubits[5], THz radiation sources[6], or resonator cavities for particle accelerators[7]. Yet, little is known about what happens to a vortex driven by very strong currents at the depairing limit $J \sim J_d$ and what is the maximal terminal velocity a vortex can reach. Moreover, even a more fundamental question of whether the notion of a moving vortex as a stable topological defect[8, 9] remains applicable at ultrahigh velocities has not been explored.

At high current densities with $J \gg J_c$ the effect of the disorder-induced pinning force diminishes and the resulting velocity of a vortex $v$ is mainly determined by the balance of the driving Lorentz force per vortex unit length $F_L = \phi_0 J$ and the viscous drag force, $F_d = \eta(v)v$[8–10]. At small $v$ the viscous Bardeen-Stephen drag coefficient, $\eta_0 \simeq \phi_0^2/2\pi\xi^2\rho_n$, results from dissipation in a circular, non-superconducting vortex core of radius $\simeq\xi$. Here, the coherence length $\xi = \hbar v_F/\pi\Delta$ defines the size of the Cooper pair in the clean limit, $\Delta$ is the superconducting gap at $J = 0$, $v_F$ is the Fermi velocity, $\rho_n$ is the normal state electrical resistivity, and $\phi_0 = 2.07 \times 10^{-15}$ Wb is the magnetic flux quantum[8, 9]. Since the current density is limited by the depairing value $J_d \simeq \phi_0/4\pi\mu_0\lambda^2\xi$ at which the speed of the superconducting condensate reaches the pair-breaking velocity $v_{dp} = \Delta/mv_F = \hbar/\pi m\xi$[9], the maximal vortex velocity can be extrapolated to $v_c \simeq \phi_0 J_d/\eta_0 \simeq \rho_n\xi/2\mu_0\lambda^2$, where $\lambda$ is the magnetic penetration depth and $m$ is the effective electron mass[7]. For a Pb film with $\lambda = 96$ nm and $\xi = 46$ nm at $T = 4.2$ K, and $\rho_n \sim 20$ nΩm[11], we obtain $v_{dp} \simeq 0.4$ km/s and $v_c \sim 40$ km/s, which suggests that the vortex could move at a velocity that is two orders of magnitude higher than the maximal drift velocity of the Cooper-pair condensate. A vortex moving much faster than the perpendicular current superflow which drives it raises many fundamental issues. What is the maximal terminal

velocity that a single vortex can actually reach and what are the mechanisms that set this limit? Does a vortex remain a well-defined topological defect even under the extreme conditions of the strongest possible current drive? Does the superfast vortex matter form dynamic patterns qualitatively different from the conventional flux flow at low velocities? Some of these issues have been studied in numerical simulations of the time-dependent Ginzburg-Landau (TDGL) equations, which, however, are only applicable at temperatures very close to $T_c$[12–16]. Since suitable theoretical frameworks for exploring the extreme dynamics of superfast vortices at low temperature have not yet been developed (Supplementary Note 1), the role of the experiment becomes paramount.

Addressing the physics of fast vortices experimentally is extremely challenging. For instance, inferring the terminal velocity $v_c$ from the conventional measurements of dc voltage–current ($V–I$) characteristics[16–19] is rather indirect because it assumes that all vortices move with the same constant velocity, which is not the case, as will be shown below. Therefore, a local probe capable of tracing vortices moving at supersonic velocities is required. A number of methods, including STM[20–22], MFM[23, 24], magneto-optical imaging[25], scanning superconducting quantum interference device (SQUID) microscopy[26], and scanning Hall probes[27–29] have been employed to image slowly moving vortex structures, but none of them could resolve the properties of high-speed vortices. In this work, we employ a novel SQUID that resides on the apex of a sharp tip[30] and provides high spatial-resolution magnetic imaging[31, 32] reaching single-spin sensitivity[30] and enabling detection of sub-nanometer ac vortex displacements[11]. Using this nanoscale SQUID-on-tip (SOT), we report the first direct microscopic imaging of superfast vortices under current densities approaching the depairing limit. Our experiment revealed vortex velocities up to tens of km/s, cascades of striking branching instabilities, and dynamic transitions in the moving vortex matter. Comprehension of the fundamental vortex properties under these extreme, previously unexplored conditions is essential for reducing dissipation and preventing breakdown of superconductivity in high current applications.

## Results

**Imaging of stationary and flowing vortices.** Our experiments were performed on a Pb film with thickness $d = 75$ nm and $T_c = 7.2$ K patterned into a 10 μm-wide microbridge with a

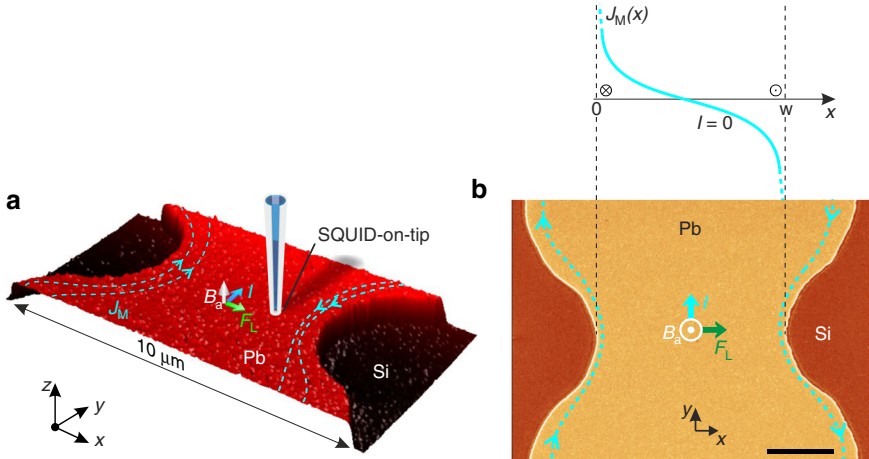

**Fig. 1** Pb thin film sample and the experimental set-up. **a** 3D representation of a $10 \times 5$ μm² AFM scan of the sample of 75 nm-thick Pb film patterned into a 10 μm-wide strip with a 5.7 μm wide constriction. Indicated are the directions of the applied magnetic field $B_a$, current $I$, the Lorentz force acting on vortices $F_L$, and the screening (Meissner) current density $J_M$ that is maximal along the edges. **b** SEM image of the same sample with corresponding distribution of the Meissner current $J_M(x)$ across the construction in absence of vortices and applied current. Scale bar is 2 μm

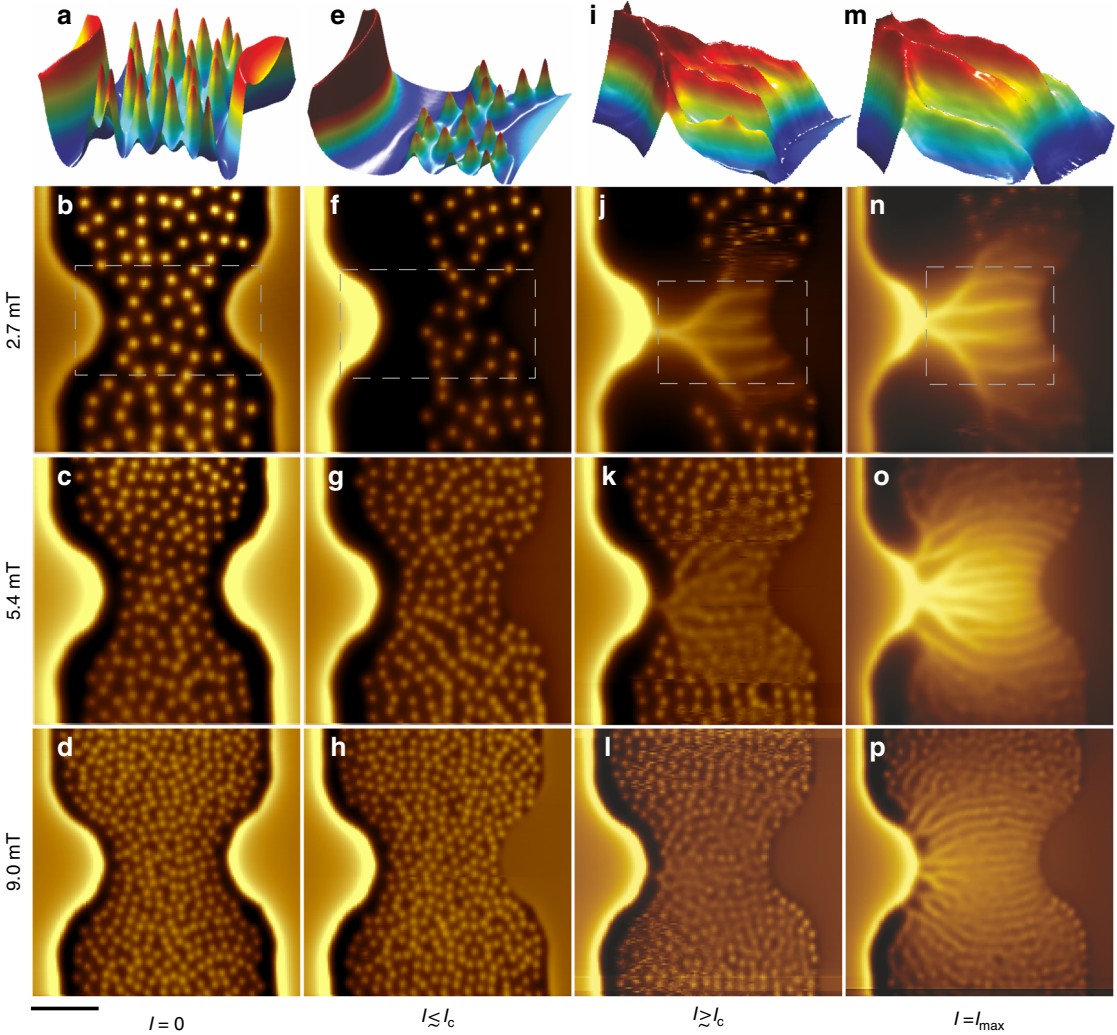

**Fig. 2** Magnetic imaging of stationary and fast moving vortices in Pb film at 4.2 K. **a–d** $B_z(x, y)$ SQUID-on-tip images of vortex configurations at $I = 0$ for different values of applied field $B_a = 2.7$ **a,b**, 5.4 **c**, and 9.0 mT **d**. **e–h** Images acquired at the verge of vortex motion at $I \lesssim I_c$, at $B_a = 2.7$ mT and $I = 16$ mA **e**, **f**, $B_a = 5.4$ mT, $I = 12.2$ mA **g**, and $B_a = 9.0$ mT, $I = 6.0$ mA **h**. **i–l** Images of onset of vortex flow at $I \gtrsim I_c$ at $B_a = 2.7$ mT, $I = 18.9$ mA **i,j**, $B_a = 5.4$ mT, $I = 12.4$ mA **k**, and $B_a = 9.0$ mT, $I = 9.1$ mA **l**. **m–p** Vortex flow patterns at the highest sustainable current with $B_a = 2.7$ mT, $I = 20.9$ mA **m,n**, $B_a = 5.4$ mA **o**, and $B_a = 9.0$ mT, $I = 11.8$ mA **p**. The color scale represents the out-of-plane field $B_z(x, y)$ with span of 1.8 **b**, 2.5 **c**, 3.0 **d**, 2.9 **f**, 3.2 **g**, 3.4 **h**, 3.1 **j**, 3.4 **k**, 3.4 **l**, 3.1 **n**, 3.6 **o**, and 2.8 mT **p**. All 2D images are $12 \times 12$ μm$^2$, pixel size 40 nm, and acquisition time 240 s/image. The scale bar is 3 μm. The top row shows zoomed-in 3D representation of $B_z(x, y)$ in the corresponding *dashed areas* marked in the second row. Supplementary Movies 1–4 for full set of images

central constriction of width $w = 5.7$ μm (Fig. 1, Supplementary Note 2 and Supplementary Figs 1 and 2). In this geometry vortices only penetrate in the narrowest part of the bridge, which greatly reduces heating. Imaging of the local magnetic field $B_z(x, y)$ above the film surface at 4.2 K was done using a 228 nm diameter SOT incorporated into a scanning probe microscope (see Methods section). Figure 2a–d shows the distribution of vortices in the strip after field cooling in $B_a = 2.7$, 5.4, and 9.0 mT, which display a disordered vortex structure pinned by material defects. The observed vortex density is not uniform, as one may expect under field cooling conditions, but has a dome-shaped profile with a maximum in the center surrounded by vortex-free stripes along the edges. This is the result of the geometrical barrier[33] which is strikingly demonstrated here with single-vortex resolution (Supplementary Fig. 3). Unlike a bulk-superconductor in which the screening currents flow in a narrow layer of thickness $\lambda$ at the surface, in a thin film strip of width $0 < x < w$ and thickness $d \ll w$ in perpendicular field $B_a$, the shielding current density in the Meissner state $J_M(x) = B_a(w - 2x)/(d\mu_0\sqrt{x(w - x)})$ varies over much larger scales and

decreases slowly $J_M(x) \propto x^{-1/2}$ and $J_M(x) \propto -(w - x)^{-1/2}$ away from the left and right edges, respectively (Fig. 1b). These currents push vortices into the central part of the strip, where they form a magnetic flux dome surrounded by vortex-free regions[33–35]. The vortex-free region shrinks with $B_a$ as seen in Fig. 2.

These vortex-free regions have a major effect on vortex dynamics in the presence of transport current. As the applied current $I$ is increased, the sum of transport and shielding current densities $J(x)$ increases at the left edge ($x = 0$) and decreases at the right edge ($x = w$). As a result, the vortex dome shifts toward the right edge and the vortex-free region at the left edge expands[33, 36], as shown in Fig. 2e–h at $I \lesssim I_c$. At the critical current $I = I_c$, the current density at the left edge approaches the depairing limit $J(0) \cong J_d$, and the flux dome reaches the right edge where $J(x)$ vanishes, so that the conditions for the onset of vortex motion are met. Here, the critical state revealed with a single-vortex resolution, is dominated by the geometrical and extended surface barriers[33–37], which has two essential differences as compared to the continuum, pinning-dominated Bean critical state[38, 39]. First, unlike the Bean state in which the vortex density is highest at the

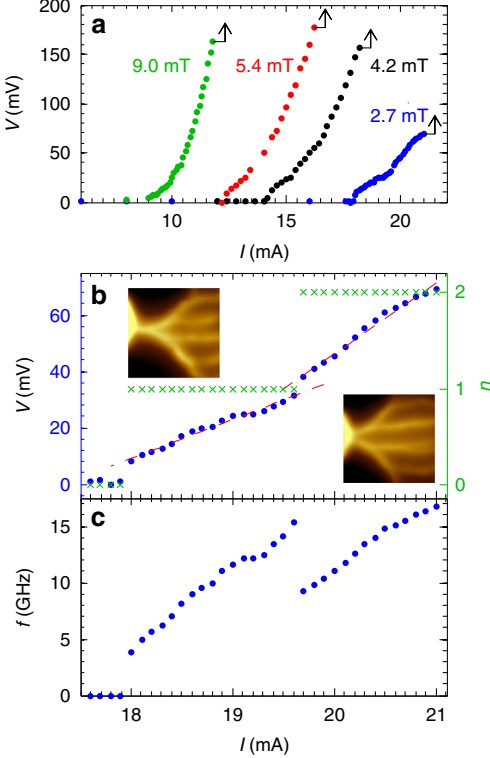

**Fig. 3** Current-voltage-frequency characterization of vortex penetration. **a** Voltage $V$ across the microbridge as a function of current $I$ for various indicated fields. **b** Voltage across the bridge (*blue*) and the number of vortex stems $n$ (*green*) as a function of current at $B_a = 2.7$ mT. The *red dashed lines* are linear fits with $dV/dI = 13.9$ mΩ in the single stem and 25.1 mΩ in double stem regions. The insets show zoomed-in SOT images of single stem and double stem vortex flow. **c** Vortex penetration rate $f$ per stem vs. current $I$

penetration edge of the sample, Fig. 2e–h shows zero vortex density at the penetration side (*left*). Secondly and most importantly, in the Bean model at $I = I_c$ the current density equals $J_c$ across the entire sample, whereas our thin film bridge is separated into two distinct regions clearly seen in Fig. 2f. In the left vortex free region, $J$ significantly exceeds the critical current density, $J_c < J < J_d$, and no stationary vortices can be present[33–36]. In the right half where $0 < J \leq J_c$ vortices are pinned. It is this unique inhomogeneous current state which allows us to investigate dynamics of superfast vortices driven by high local current densities that cannot be done by global transport measurement in bulk samples. Here the penetrating vortices can be subjected locally to extremely high current densities $J \gg J_c$ at the edges while the net current is only slightly above the critical, $I \gtrsim I_c$ and heating is weak.

As $I$ exceeds $I_c$, vortices start penetrating through the left edge of the constriction, as shown in Fig. 2i–l. Since vortices traverse the sample in about 1 ns, much shorter than our imaging time (~4 min/image), the observed images represent time-averaged locations of vortices. Remarkably, rather than entering randomly at the edge and flowing across the film while avoiding each other due to repulsive interactions, vortices penetrate at a well-defined point in the narrowest part of the bridge and follow each other forming a single channel or stem, which then undergoes subsequent bifurcations. The number of stems and the number of branches increases with increasing $I$ and $B_a$, as shown in Fig. 2i–p (Supplementary Movies 1–4). The intensity of the local $B_z(x, y)$ along the channels is proportional to the fraction of time

spent by a vortex at a specific location. The variations in $B_z(x, y)$ with distinct maxima along some of the channels (e.g., Fig. 2i) thus reveal the locations where the vortices slow down due to pinning and vortex–vortex interlocking. The field profiles along the channels become more uniform as the current increases (Fig. 2m).

**Transport properties and vortex penetration frequency.** In order to extract the vortex velocity, we measured the $V–I$ characteristics simultaneously with the SOT imaging, as shown in Fig. 3a. At each field, the onset of finite voltage $V$ coincides with the appearance of the first vortex channel as $I$ exceeds a critical current $I_c(B_a)$ that decreases with $B_a$. The data points end at the maximal current above which the voltage jumps abruptly by more than an order of magnitude (*black arrows*) apparently due to a thermal quench in the constriction region. Such simultaneous SOT and transport measurements provide a unique opportunity to reveal superfast dynamics of vortices and their trajectories with a single-vortex resolution. Figure 3b shows the measured voltage drop on the bridge (*left axis*) along with the number of vortex stems $n$ observed by SOT imaging (*right axis*) vs. current at $B_a = 2.7$ mT (see Supplementary Movie 1). The appearance of each subsequent stem in Fig. 3b matches a step in the voltage and a change in the differential resistance $dV/dI$. Linear fits to the data (*dashed*) show a roughly twofold increase in $dV/dI$, from 13.9 mΩ for one stem to 25.1 mΩ for two stems. For a given number of stems $n$, the vortex penetration frequency $f$ in each stem is given by the Faraday law, $f = V/n\phi_0$. Figure 3c shows that the penetration frequency jumps from zero to 3.7 GHz at the formation of the first stem. As $I$ increases, $f$ rises to 15.3 GHz and then drops abruptly to 9.1 GHz upon the formation of the second stem.

**Vortex velocity.** Vortex conservation requires that $f = V/\phi_0$ remains constant along the stem up to the bifurcation point, so that the vortex velocity along the stem is given by $v(x) = fa(x)$, where $a(x)$ is the local average intervortex distance. Our simultaneous SOT and transport measurements allow determination of $a(x)$ and $v(x)$ as follows. The average field along a chain of vortices separated by $a(x)$ is given by $B_{av}(x) = \int_{-\infty}^{\infty} B_v(u) du/a(x)$, where $B_v(u)$ is the magnetic field profile of an individual vortex. By measuring $B_{av}(x)$ along the stem and $B_v(x)$ across an isolated stationary vortex (Supplementary Note 3 and Supplementary Fig. 4), we thus obtain $a(x)$ along a single stem (Fig. 4a) from $x = 0.5$ μm up to the bifurcation point $x = x_b$, for various currents at $B_a = 2.7$ mT. Taking the penetration rate $f$ from Fig. 3c, we derive the corresponding vortex velocity $v(x) = fa(x)$ as shown in Fig. 4b. The resulting $v(x)$ decreases with the distance $x$ from the edge, consistent with the decreasing current density near the edge[33–36], $J(x) \simeq J_d(\Lambda/x)^{1/2}$, which drives the vortices, where $\Lambda = 2\lambda^2/d$. The remarkable findings here are the extreme values of vortex velocities of 10–20 km/s, much larger than the depairing superfluid velocity $v_d \simeq 0.4$ km/s estimated above.

These very high velocities are attained at 0.5 μm $< x <$ 1.5 μm where the estimated current density is $0.6J_d > J > 0.2J_d$. In the region of $0 < x < 0.5$ μm at the film edge, where even higher currents and vortex velocities are possible, our SOT technique cannot resolve the actual $v(x)$ as the vortex field $B_v(x)$ close to the edge becomes partly extinguished by the image vortex imposed by the boundary conditions[40] (Supplementary Note 3). Figure 4c shows that the vortex velocities at $x = 0.5$ μm and at $x = x_b$ increase monotonically with $I$, whereas the length of the stem defined by the bifurcation point $x_b$ decreases with $I$, as seen in Fig. 4b.

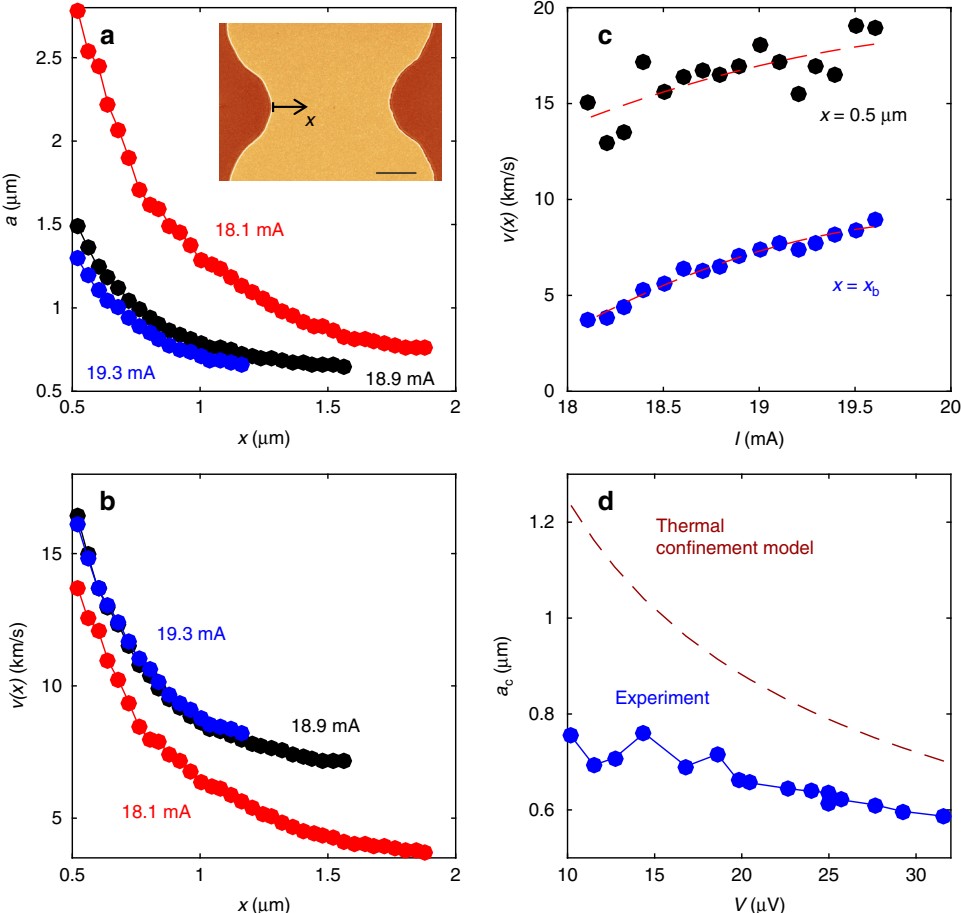

**Fig. 4** Vortex velocities and spacing along the entrance stem channel. **a** Spacing between successive vortices $a(x)$ along the stem from $x = 0.5\,\mu m$ up to the bifurcation point $x_b$ at $B_a \cong 2.7\,mT$ at various indicated currents. Inset: SEM image of the sample with marked $x$ axis. The scale bar is $2\,\mu m$. **b** Corresponding vortex velocities $v(x)$ along the stem from $x = 0.5\,\mu m$ up to the bifurcation point $x_b$. **c** The vortex velocity vs. current at $x = 0.5\,\mu m$ (*black*) and at $x_b$ (*blue*). The *dashed lines* are guides to the eye. **d** Vortex spacing $a_c$ at the bifurcation point $x_b$ vs. the voltage $V$ across the microbridge (*blue*), compared to the theoretical estimate (*dashed*) based on the thermal confinement model

The average vortex velocity in the stem can be estimated independently of the above analysis by assuming the distance between the moving vortices to be of the order of their mean stationary distance $a \cong 1\,\mu m$ from Fig. 2b (which is close to $a = \left(2\phi_0/\sqrt{3}B_a\right)^{1/2} = 0.94\,\mu m$) and taking the highest frequency $f \cong 15\,GHz$ from Fig. 3c. This yields $v = fa \cong 15\,km/s$ which is consistent with the measured vortex velocities in Fig. 4b.

The mesoscopic chains of single vortices moving along stationary channels under a dc drive and weak overheating reported here are fundamentally different from transient dendritic flux avalanches observed by magneto-optical imaging in increasing magnetic fields[41–45]. Those macroscopic filaments of magnetic flux focused in regions overheated above $T_c$ can propagate with velocities as high as $150\,km/s$ in $YBa_2Cu_3O_7$ films at $10\,K$[43] or $360\,km/s$ in $YNi_2B_2C$ at $4.6\,K$[44]. Such thermomagnetic avalanches are driven not by the motion of single vortices but by strong inductive overheating caused by the fast-propagating stray electromagnetic fields outside the film[45], unlike the correlated flow of quantized vortices reveled by our SOT imaging under nearly isothermal conditions. The mechanisms of channeling and branching of fast vortices in our viscosity-dominated regime at $J \gg J_c$ are also different from the disorder-driven formation of networks of slower vortices near the depinning transition observed in numerical simulations[46, 47].

## Discussion

Our experimental findings raise many fundamental questions: What are the mechanisms of vortex confinement in the channels? Why does the branching instability occur and what controls the number of vortex stems? What are the mechanisms which determine the terminal velocities of vortices? How does the structure of a vortex evolve at the superfast velocities observed in our experiments? Given the lack of theory to describe vortices under such extreme conditions, we only limit ourselves to a qualitative discussion of essential effects which may help understand our SOT observations.

The stationary pattern of vortex channels shown in Fig. 2 seems counterintuitive, since vortices repel each other and should therefore disperse over the film. Moreover, each stem grows into a tree through a series of subsequent bifurcations but the branches of different trees do not merge. In order to keep vortices within each channel a mechanism for dynamic alignment of fast moving vortices must be present. One such mechanism is that a rapidly moving vortex leaves behind a wake of reduced order parameter which attracts the following vortex. As a result, a confined chain of vortices in a self-induced channel of reduced superfluid density can be formed (Supplementary Note 1), as it was observed previously in numerical TDGL simulations[15, 16] at high currents, $J \sim J_d$, and $T \approx T_c$. As discussed below, this mechanism apparently becomes dominant at velocities substantially higher than those accessible in our experiment.

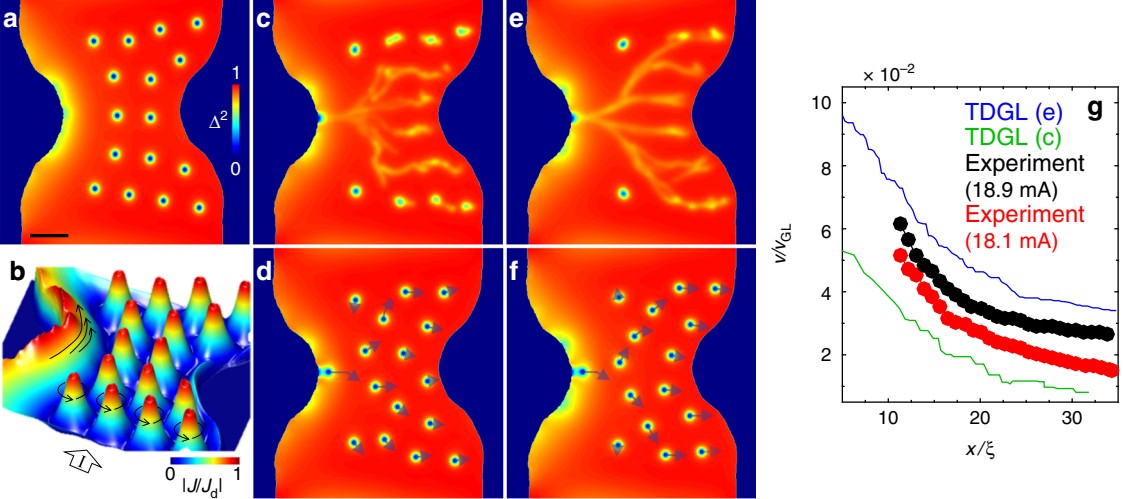

**Fig. 5** TDGL simulations of stationary and fast moving vortices at the experimentally accessible velocities. **a** Calculated Cooper-pair density $\Delta^2(x, y)$ of a stationary vortex configuration at applied current density and magnetic field corresponding to the experimental conditions in Fig. 2f. **b** Corresponding distribution of the supercurrent density $|J(x, y)|/J_d$ in the sample showing edge currents in the constriction reaching $J_d$ at the verge of vortex penetration. The *black arrows point* to the local direction of the current. **c** Time-average of the Cooper-pair density over $5 \times 10^4 \tau_{GL}$ at $I = 1.05 I_c$, revealing branching vortex trajectories coexisting with adjacent stationary vortices. **d** Snapshot of moving vortices in **c** with arrows denoting the relative displacement of each vortex following an entry of a new vortex into the sample. **e,f** Same as **c,d** but at highest applied current before an additional stem is formed. **g** Experimental vortex velocity along the stem for $B_a = 2.7$ mT and indicated applied currents with the TDGL data from **d** and **f** in normalized units (scaled to $v_{GL} = \xi/\tau_{GL}$). The animation of the vortex flow dynamics corresponding to **e,f** is presented in Supplementary Videos 5 and 6. The scale bar in **a** is 20 $\xi$

Vortex alignment may also result from a weak quasiparticle overheating: the power $P = \eta v^2$ generated by each vortex produces a channel of enhanced temperature along the moving vortex chain. The resulting bell-shaped temperature distribution $T(y)$ across the channel causes a restoring force $f_r = -s^* dT/dy$ which stabilizes the vortex chain against buckling distortions (Supplementary Note 4), where $s^*(T)$ is the transport entropy that defines thermoelectric effects in superconductors[7]. The thermal confinement resulting in long-range alignment of vortices is particularly effective at large vortex spacing $a \gg \xi$ relevant to $a \simeq$ 1–2 μm. The observed spacing along the stem varies from 0.6 to 2 μm (Fig. 4a) much larger than $\xi = 46$ nm, which suggests that the thermal confinement is dominant. As shown in Supplementary Note 4, this mechanism can be effective, even at weak overheating, without causing any thermo-magnetic instabilities.

Once a confined channel is formed, why would it bifurcate? A chain of aligned vortices can become unstable and buckle as repelling vortices get closer to each other. In a thin film, the long-range surface currents produced by vortices result in the repulsion force $f_m = \phi_0^2/\mu_0 \pi a^2$ between vortices spaced by $a \gg 2\lambda^2/d \simeq 250$ nm[8]. As the vortex spacing drops below a critical value $a_c$, the net repulsion force pushing a vortex displaced by $u$ perpendicular to the channel, $f_\perp \sim u\phi_0^2/\mu_0 a^3$, exceeds the restoring force $f_r = -ku$, leading to chain bifurcation, where the spring constant $k(a)$ is determined by the confinement mechanism. The thermal confinement leads to the critical spacing that decreases with the voltage $V$ across the channel as $a_c \propto V^{-1/2}$ (Supplementary Note 4), in qualitative agreement with the experimental data shown in Fig. 4d. As described in Supplementary Note 5, transverse displacements of fast vortices at $J \gg J_c$ can be enhanced by the weak effect of disorder which may cause premature bifurcation of vortex channels.

To gain an insight into the structure and the viscous dynamics and channeling of fast vortices, we performed numerical simulations of TDGL equations for the bridge geometry of Fig. 1 and the material parameters of our Pb films (see Methods section). We limit ourselves to a minimal model of superfast vortices driven by strong current densities $J \gg J_c$ for which disorder and heating was

neglected. The simulated Cooper pair density $\Delta^2(x, y)$ shown in Fig. 5a reproduces the main features of the SOT images at $I \lesssim I_c$, namely vortices displaced to the right edge and a pronounced vortex-free region along the left edge. Notice that in the absence of disorder, the stationary vortices in Fig. 5a form an ordered structure within a smooth confining potential of the geometrical barrier, in contrast to Fig. 2f which shows a disordered vortex configuration determined by pinning in the right-hand side of the sample where $J < J_c$. At $I = I_c$ the calculated current density $J(x)$ shown in Fig. 5b reaches the depairing limit $J_d$ at the left edge of the constriction and vanishes at the opposite edge.

At $I > I_c$ vortices start penetrating through the left edge and move along a network of preferable paths forming a branching tree with an overall shape determined by the bridge geometry. The vortex chains are curved on larger scales (Fig. 5e) due to the lensing effect of the current distribution in the constriction, which tends to orient the vortex chains perpendicular to the local current $J(x, y)$. The calculated vortex flow pattern is similar to the SOT image in Fig. 2j and also exhibits the coexistence of moving and stationary vortices as observed in Fig. 2 where bulk pinning further hampers the motion of remote vortices. The Copper pair density averaged over the simulated period of time shows a non-uniform distribution along the vortex channels (Fig. 5c, e) with distinct bright spots of reduced Cooper pair density indicating that the vortex velocity varies non-monotonically along the channels. The bright spots describe the regions where the vortices slow down or even stop momentarily, giving rise to vortex crowding (Supplementary Movies 5–8). Similar features of $B_z(x, y)$ along the channels are observed in Fig. 2i–p. Such vortex "traffic jams" can be understood as follows. A vortex penetrating from the left edge slows down as it moves along the channel since the driving current $J(x, y)$ decreases across the strip. The subsequent penetrating vortices move along the same trajectory, causing jamming in the regions where vortices slow down. The resulting mutual repulsion of vortices either pushes them further along the channel, where vortices speed up due to attraction to the right edge of the strip, or causes bifurcation of the channel into branches.

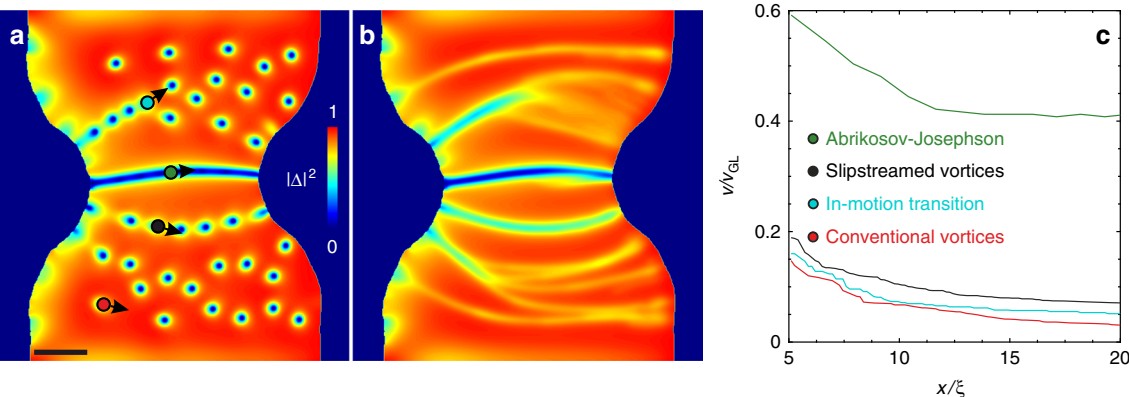

**Fig. 6** Different morphologies of ultra-fast vortices at velocities significantly higher than in our experiment. **a,b** A snapshot **a** and time-averaged Cooper-pair density $\Delta^2(x, y)$ **b** as in Fig. 5, but for twice higher applied field and twice the current. Three vortex phases are found with distinctly different core structure, level of quasiparticle tailgating, velocities and resulting kinematic trajectories (see text and Supplementary Movies 7 and 8), namely the extremely fast Abrikosov-Josephson vortices (marked by *green dot*), the ultrafast slipstreamed vortices (*black dot*), and conventional Abrikosov moving vortices (*red dot*). **c** Spatial profiles of vortex velocities $v(x)$ (scaled to $v_{GL} = \xi/\tau_{GL}$, see Methods) for the three main vortex phases, and for one detected branch of vortices going through an in-motion transition (dynamic transition from slipstreamed vortices to conventional Abrikosov vortices, identified by a *blue dot* in **a**). The scale bar in **a** is 20 $\xi$

The fact that there is a single vortex entry point and several exit points necessitates that the penetration frequency per stem is higher than the exit frequency per channel. Figure 5d, f shows snapshots of vortex motion at different applied currents, with arrows proportional to the instantaneous velocities of vortices right after penetration of a new vortex. We find that the periodically-entering vortices take alternating routes at the bifurcation points due to interactions with other vortices, which slow down further after the bifurcation (Supplementary Movies 5–8).

Figure 5g presents the calculated vortex velocity $v(x)$ along the stems in Fig. 5d, f juxtaposed with the experimental data. The decreasing vortex velocity follows the drop in $J(x)$ from the left edge of the constriction, $v(x) = \phi_0 J(x)/\eta$, showing no significant velocity dependence of $\eta(v)$ at these values of $v$. This allows us to extrapolate the experimental $v(x)$ to $x \simeq \xi$ at the entrance point, where the velocity is maximal, $v(\xi) \simeq J_d\phi_0/\eta = 24$ km/s, assuming a constant $\eta(v)$.

Using the current density $J(x)$ obtained from the simulations and $v(x)$ extracted from our experimental data, we obtain the vortex viscous drag coefficient $\eta = 2.6 \times 10^{-8}$ kgm$^{-1}$ s$^{-1}$. This value of $\eta$ is of the order of $\eta_0 \simeq \phi_0^2/2\pi\xi^2\rho_n = 10^{-8}$ kgm$^{-1}$ s$^{-1}$ of the Bardeen-Stephen model, which indicates no excessive changes in the structure of the Abrikosov vortex core even at velocities of the order of 10 km/s. This conclusion is corroborated by our TDGL simulations in Fig. 5 which reproduce the channel bifurcations due to vortex repulsion and the variation of $B_z(x, y)$ along the channels due to disorder and interactions induced variations in vortex velocity. The totality of our SOT and TDGL results indicate that vortices maintain their integrity as stable topological defects even at the observed extreme velocities for which the magnetic field of a moving vortex does not deviate substantially from that of a stationary Abrikosov vortex. In particular, we have observed no evidence of the transition of Abrikosov vortices into Josephson-like phase slip lines[48, 49] extending across the bridge.

Now we turn to the numerical study of even faster vortices, beyond our experimentally accessible range of parameters, for which a significant change in the internal vortex structure is expected. For instance, nonequilibrium effects can give rise to a velocity dependence of $\eta(v)$ and to the Larkin–Ovchinnikov (LO) instability caused by diffusion of quasiparticles from the vortex core[50]. The LO instability results in jumps in the vortex velocity above $J > J_{LO} \simeq \eta_0 v_0/2\phi_0$ for which the force balance $\phi_0 J = \eta(v)v$ at $v > v_0$ is not satisfied because $\eta(v) = \eta_0/(1 + v^2/v_0^2)$ decreases with $v$[50]. The LO or overheating instabilities[51, 52], have been observed on various superconductors with $v_0$ ranging from 1 to 10 km/s[17–19].

Our TDGL calculations at twice higher current and field as compared to those shown in Fig. 5, reveal three different types of vortices described in Fig. 6. Far from the constriction region, $J$ is lower and the moving vortices (*red dot* in Fig. 6a) have a regular, nearly isotropic shape with no wake of reduced order parameter. Closer to the constriction, a chain of vortices (marked by a black dot in Fig. 6a) is confined in a channel of reduced order parameter. These faster-moving vortices are slipstreaming one another because their velocity $v$ exceeds $a/\tau_\Delta$, where $\tau_\Delta = \pi\hbar\sqrt{1 + 4\tau_{in}^2\Delta^2/\hbar^2}/8k_B(T_c - T)$ is a recovery time of the superconducting order parameter in the wake of the moving vortex and $\tau_{in}$ is an electron-phonon inelastic scattering time. Our TDGL simulations show that these vortices, moving in channels, have elongated cores along the direction of motion, and their drag coefficient can be approximated by the LO dependence $\eta_{LO}(v) = \eta_0/(1 + v^2/v_0^2) + \eta_i$ with $\eta_i \approx 0.25\eta_0$ and $v_0 \approx \xi/\tau_\Delta \approx 20$ km/s for our sample parameters (see Methods section). These anisotropic slipstreamed vortices can undergo a kinematic transition to conventional vortices upon stem bifurcation which leads to additional vortex slowdown, as marked by a *blue dot* in Fig. 6a.

The most radical change in the structure of moving vortices occurs in the narrowest part of the constriction, where $J$ is maximal. Here a channel with a significant reduction of the mean superfluid density appears, in which ultrafast vortices (*green dot* in Fig. 6a) are moving with velocities that are 3–5 times higher than the speed of slipstreamed vortices, as shown in Fig. 6c. The ultrafast vortices in the central channel can be regarded as Josephson or mixed Abrikosov-Josephson vortices similar to vortices at grain boundaries[53, 54], in high-critical-current planar junctions[55], or S/S′/S weak links[56]. The TDGL results shown in Fig. 6c suggest that Josephson-like vortices in these channels can move with velocities as high as ~100 km/s, because the viscous drag coefficient $\eta(v)$ in the channel is reduced to just a few percent of $\eta_0$ due to strongly elongated and overlapping vortex cores. Spatial modulation of the order parameter between these vortices is rather weak and effectively the channel behaves as a

self-induced Josephson junction, which appears without materials weak links. Similar flux channels in thin films were previously interpreted in terms of phase slip lines[48, 49]. In the case of strong suppression of the order parameter and weak repulsion of Josephson vortices which extend over lengths exceeding $\Lambda = 2\lambda^2/d$, the channel does not bifurcate as shown in Fig. 6a and the magnetic field along the channel is nearly uniform. This feature of Josephson-like vortices appears inconsistent with our SOT observations of vortex channels which always bifurcate and show noticeable variations of $B_z(x, y)$ along the channels. The SOT results thus indicate an essential effect of intervortex repulsions and weak suppression of the order parameter along the channel, consistent with the dynamics of Abrikosov vortices shown in Fig. 5.

Another interesting SOT observation shown in Fig. 2n–p is the nucleation of additional stems of vortices as current increases. This effect can be understood as follows. The first stem appears at $I = I_c$ as the local current density $J_s$ at the edge of the constriction reaches $J_d$. As $I$ increases above $I_c$, vortices start penetrating at the narrowest part of the constriction in such a way that a counter-flow of circulating currents produced by a chain of vortices moving in the central channel maintains the current density $J_s(y) < J_d$ everywhere along the curved edge of the film except for the vortex entry point. This condition defines the spacing $a(I)$ between the vortices in the chain. However, above a certain current $I > I_1$, a single chain of vortices can no longer maintain $J_s(y) < J_d$ along the rest of the constriction edge, leading to nucleation of an additional stem as seen in Fig. 2n. Our calculation presented in Supplementary Note 6 and Supplementary Figs 5 and 6 show that the second stem appears at the current $I_1 = I_c \left[ 1 + \left( 5\sqrt{3}\xi/d \right)^{1/2} \lambda/R \right]$ which depends on the radius of curvature $R$ of the constriction. For $\xi = 46$ nm, $d = 75$ nm, $\lambda = 96$ nm, and $R = 2$ μm, we obtain $I_1 = 1.11 I_c$ in good agreement with the observed $I_1 = 1.09 I_c$ at $B_a = 2.7$ mT in Fig. 3b. In addition, the edge roughness can affect the location and the dynamics of stem evolution, favoring stem nucleation at points of local edge protuberances (Supplementary Note 6). We have incorporated the actual details of the edge shape of our sample derived from the SEM image into our TDGL simulations resulting in the observed asymmetry between the vortex channels in the upper and lower parts of Fig. 6a, b.

As the magnetic field increases, the width of the vortex-free region near the edges and vortex velocities decrease. Figure 2 shows how dissipative vortex structures evolve from a few mesoscopic chains and branches sustaining extremely high vortex velocities at low field (Fig. 2j, n) to a multi-chain structure with much lower vortex velocities at higher fields (Fig. 2l, p). Remarkably, the vortex channeling is preserved even at high fields that would usually be associated with the conventional flux flow of an Abrikosov lattice. The dynamic structure revealed in Fig. 2p, in which vortices move in parallel channels, appears consistent with the predictions of the moving Bragg glass theory[57], thus providing microscopic evidence with a single vortex resolution for the existence of this dynamic phase.

In conclusion, this work uncovers the rich physics of ultrafast vortices in superconducting films and offers a broad outlook for further experimental and theoretical investigations. By proper sample design and improved heat removal it should be possible to reach even higher velocities for investigation of non-equilibrium instabilities[17, 18, 50–52, 58]. Our detection of vortices moving at velocities of up to 20 km/s, significantly faster than previously reported, strengthens the recently renewed appeal of vortex-based cryogenic electronics[59]. The observed frequencies of penetration of vortices in excess of 10 GHz may be pushed to the much technologically desired THz gap in the case of dynamic Abrikosov-Josephson vortex phases. This work shows that the

SOT technique can address some outstanding problems of nonequilibrium superconductivity and ultrafast vortices in type II superconductors as well as dynamics of the intermediate state in type I superconductors on the nanoscale. These issues can also be essential for further development of superconducting electronics, opening new challenges for theories and experiments in the yet unexplored range of very high electromagnetic fields and currents.

## Methods

**Scanning SQUID-on-tip microscopy.** The SOT was fabricated using self-aligned three-step thermal deposition of Pb at cryogenic temperatures, as described previously[30, 60, 61]. Supplementary Figure 7 shows the measured quantum interference pattern of the SOT used for this work with an effective diameter of 228 nm, 135 μA critical current at zero field, and white flux noise down to 270 n$\phi_0$Hz$^{-1/2}$ at frequencies above a few hundred Hz. The slightly asymmetric structure of the SOT gives rise to a shift of the interference pattern resulting in good sensitivity even at zero applied field with flux noise of 1.6 μ$\phi_0$Hz$^{-1/2}$. All the measurements were performed at 4.2 K in He exchange gas of ~1 mbar.

**Sample fabrication.** A 75 nm-thick Pb film was deposited by thermal evaporation onto a Si substrate cooled to liquid nitrogen temperature in order to reduce the high surface mobility of Pb atoms and limit island growth. The base pressure was $2.2 \times 10^{-7}$ Torr and the deposition rate was 0.6 nm/s. A protective layer of 7 nm of Ge was deposited in situ to prevent oxidation of the Pb. The sample was patterned using a standard lift-off lithographic process. The film was characterized by Scanning Electron Microscopy (SEM) and Atomic Force Microscopy (AFM) as described in Supplementary Note 2 and Supplementary Figs. 1 and 2. Further characterization of films grown under the same conditions are described in ref. [11].

**Vortex state preparation.** To prepare the initial field-cooled vortex state a current of a few tens of mA was applied to the microbridge, heating it to above $T_c$. The current was then turned off and the film was field-cooled in the desired applied magnetic field. Supplementary Fig. 3 shows six $12 \times 12$ μm$^2$ scans of the microbridge field cooled in 0.3, 0.6, 1.5, 2.7, 5.4 and 12 mT.

**Numerical simulations.** The numerical simulations of the kinematic vortex states were performed using the generalized TDGL model for a gapped dirty superconductor[9], where the equation for the complex order parameter $\Psi(\mathbf{r}, t) = \Delta(\mathbf{r}, t)e^{-i\theta(\mathbf{r},t)}$,

$$\frac{u}{\sqrt{1 + \gamma^2 |\Psi|^2}} \left( \frac{\partial}{\partial t} + i\varphi + \frac{\gamma^2}{2} \frac{\partial |\Psi|^2}{\partial t} \right) \Psi = (\nabla - i\mathbf{A})^2 \Psi + \left(1 - |\Psi|^2\right)\Psi, \quad (1)$$

is solved self-consistently with the equation for the electrostatic potential $\varphi$:

$$\nabla^2 \varphi = \nabla(Im\{\Psi^*(\nabla - i\mathbf{A})\Psi\}). \quad (2)$$

These equations are given in dimensionless form. The distances are expressed in units of the coherence length $\xi$, the time in units of $\tau_{GL} = \pi\hbar/8k_B T_c(1 - T/T_c)u$, where $u = 5.79$ is given by the TDGL theory in the dirty limit[9]. The complex order parameter $\Psi$ is given in units of $\Delta_0 = 4k_B T_c u^{1/2}(1 - T/T_c)^{1/2}/\pi$, and electrostatic potential $\varphi$ in units $\varphi_0 = \hbar/e^*\tau_{GL}$. Vector potential $\mathbf{A}$ is scaled by $A_0 = \varphi_0/2\pi\xi$ and current density $j$ is given in units of $j_{GL} = \sigma_n \varphi_0/\xi$, where $\sigma_n = 1/\rho_n$ is the normal state conductivity. Parameter $\gamma = 2\tau_{in}\Delta_0/\hbar$ contains the influence of the inelastic phonon-electron scattering time $\tau_{in}$ on the dynamics of the superconducting condensate.

The simulations were implemented using a finite difference method, on a Cartesian map with a dense grid spacing of 0.1 $\xi$, where we reproduced the geometry of the experimental specimen based on the SEM image. Due to memory and time constraints of the self-consistent calculation, the actual size of the simulated sample was taken twice smaller than the experimental specimen, still making a formidable numerical effort on a $\sim 10^3 \times 10^3$ two-dimensional spatial mesh, where parameter $\gamma = 100$ was taken as an order of magnitude estimate for Pb (correspondingly lowering the time step in the used implicit Crank–Nicolson method). Equation (2) was solved by a spectral Fourier procedure. For such a demanding numerical task, we employed the GPU-based parallel computation scheme[62]. The gauge selected for the system of Eqs. 1 and 2 was $\nabla \mathbf{A} = 0$. The boundary conditions used at the superconductor-vacuum boundary were $\mathbf{n} \cdot (\nabla - i\mathbf{A})\Psi = 0$ and $\mathbf{n} \cdot \nabla\varphi = 0$ ($\mathbf{n}$ being the unit vector perpendicular to the boundary). The current was applied through normal-metal contacts sufficiently far from the constricted area, ensuring that applied current is fully transformed into normal current there ($\mathbf{n} \cdot \sigma_n\nabla\varphi = J$).

We emphasize that the TDGL theory was used here as the best, currently available, computational tool to qualitatively address the essential physics of the interplay of inhomogeneous transport and magnetization current densities which

drive the vortex matter in the bridge and produce the branching flux-flow instabilities and the striking changes in the core structure of fast moving vortices. We believe that this approach captures qualitative features of our SOT observations, although it can hardly give reliable numerical estimates of terminal vortex velocities as discussed in Supplementary Note 1. We also made more physically transparent analytic estimates of vortex confinement and buckling instabilities of vortex chains using the thermal diffusion equation and the London theory (Supplementary Notes 4–6).

**Data Availability**. The data sets generated during and/or analyzed during the current study are available from the corresponding authors on reasonable request.

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

## Acknowledgements

We would like to thank M.L. Rappaport for fruitful discussions and technical support. This work was supported by the US-Israel Binational Science Foundation (BSF) grant No. 2014155 and the Israel Science Foundation grant No. 132/14. A.G. was also supported by the United States Department of Energy under Grant No. DE-SC0010081. M.V.M. acknowledges support from Research Foundation-Flanders (FWO). The work of Ž.L.J. and A.V.S. was partially supported by "Mandat d'Impulsion Scientifique" MIS F.4527.13 of the F.R.S.-FNRS. This work benefited from the support of COST action MP-1201.

## Author contributions

E.Z., L.E. and Y.A. conceived the experiment and analyzed the data. L.E., Y.A. and E.O.L. performed the scanning SOT measurements. L.E. constructed the scanning SOT microscope. Y.A. and L.E. fabricated and characterized the samples. Y.A. and Y.M. developed the SOT fabrication technique and fabricated the SOTs. M.E.H. developed the SOT readout system. A.G., G.P.M., Ž.L.J., A.V.S., and M.V.M. carried out the theoretical analysis and the numerical simulations. A.G., Y.A., M.V.M., E.Z., L.E., and Ž.L.J. wrote the paper with contributions from all authors.

## Additional information

**Competing interests:** The authors declare no competing financial interests.

