## [Peer Review File · Nature Communications]

Reviewers' comments:

Reviewer #1 (Remarks to the Author):

I have carefully read the manuscript "Super-fast dynamics and flow instabilities of superconducting vortices" by L. Embon et al. (NCOMMS-16-25510) and I found very interesting data with important conclusive results. The manuscript deals with a powerful tool able to visualize at a nanometer scale the motion of quantized vortices driven by electric currents in superconductors. The main focus is the "imaging" of single flowing vortices at high velocities such as tens of km/s, which has not yet been achieved by other direct imaging techniques. The experimental data obtained by such a scanning SQUID-on-tip (SOT) microscopy are accompanied by numerical simulations performed using the generalized TDGL model, that has several limitations and, although qualitatively, it shows the several features of the experimental observations. Apart from the novelty, the claims are convincing and the experiment gives a direct evidence of a channel-like vortex motion into the chosen constricted geometry of the superconducting material under investigation. However there's a general comment to be done on this manuscript in order to strengthen the paper further, that is the authors should underline if the obtained results can be generalized, and how? Is the study applicable to any type of superconducting material?, whereas it seems strongly influenced by material parameters.

There are several remarks I would like to point out:

- 1) First of all, concerning the TITLE, it does not sound appropriate since it should include the keyword "Imaging", for example "Imaging of super-fast dynamics of superconducting vortices".
- 2) In the ABSTRACT the presented technique is addressed with "single spin sensitivity", although it has such peculiar characteristic, it has already been used by the authors as a "single vortex" microscopy device, as referred in the proper cited reference [10] of the manuscript.
- 3) In the Introduction part, the balancing of the driving force and the viscous one can still be done in the presence of a pinning force for $J \gg J_c$? In particular in the presence of a dynamic pinning force which is generally a velocity dependent force, see for example PRL 108, 217001 (2012). Would you comment on the two possible cases of weak and strong pinning?
- 4) The superconducting material under investigation can be considered a type-II superconductor with a very small Ginzburg-Landau parameter, does this condition influence the observed and obtained results? On the other hand Pb superconductor can easily be tuned to be type I superconductor, for which it is well known the current-induced resistive state [see e.g. "Magnetic Flux Structures in Superconductors" by R.P.Huebener edited by Springer in Solid-State Sciences], which can have several similarities with the vortex resistive state in type-II superconductors. What would you expect in this case?
- 5) In the Introduction part there's a rather strong claim on the TDGL applicability, but then this is the numerical method that is used in the following sections. The statement "TDGL equations ...are no longer applicable..." should be modified accordingly to Supplementary Note 1.
- 6) In the Introduction part there's another claim on the possibility to measure the "non constant velocity of vortices" by a local spatial probe, as it is done with this SOT technique, does it mean that the obtained results are in contrast with the results reported in the literature? On the other hand the V-I measurements have been also used by some of the authors in the reference New Journal of Physics 14 (2012) 053006.
- 7) In the experiment the field-cooled vortex state is always prepared, have you ever tried zero field cooling as the starting state? Could you show up the results on super-fast dynamics in this case too?
- 8) In Fig. 3a the V-I curves are shown for voltages up to finite values marked by black arrows.

How can these abrupt jumps be ascribed to a thermal runaway and/or not to another mechanism? What is the control of Joule self-heating during such measurements?

9) Moreover in the Discussion other mechanisms are examined, some of them are related to vortex confinement in the channels and others can be responsible of additional buckling of vortex channels itself. However it is not straightforward to follow the Discussion section, which need to be better organized and written, by clearly separating branching instabilities (a secondary effect) from vortex channeling (primary result).

10) Among the mechanisms responsible for branching instability there's the Supplementary Note 5 fully dedicated to the pinning influence, in which pinning is treated perturbatively, but in the main section of the manuscript (Discussion) it is stated that the Pb film under investigation revealed dense structure of strong pinning sites as reported elsewhere in their ref. [10]. Is it contradictory in some way?

11) Addressing the vortex structure evolution at superfast velocities, the LO dependence of the damping coefficient is assumed in the calculations, although it is claimed before in the discussion text that the LO instability has not been observed. Is it also contradictory in some way?

12) Concerning Fig.4, which shows the spatial variation of both the vortex velocity and the intervortex spacing for 1 stem, it can be inferred that the velocity become practically uniform within 1/5 of the sample constriction width for this type of geometry, I wonder if this can be generalized to wider strip geometry, in which such a spatial velocity distribution can be neglected?

13) In the Supplementary Fig.1 the six 10 micron wide microbridges with constrictions of different widths are not visible, try to have a better SEM image.

14) In the TDGL model description included in Methods there is the parameter u which is not defined, and whose value has not been justified, but it contributes to the main parameter τ_{GL} of the simulations.

15) Based on the actual granularity of the Pb film investigated, authors should comment on the effective influence of weak rather than strong pinning on the branching mechanism of vortex chains.

Finally I suggest that this manuscript deserves publication in Nature Communication, once all the criticism has been satisfactory clarified.

Reviewer #2 (Remarks to the Author):

Please see the attached PDF file with a detailed and systematic report.

This is very nice work. I like it. However, the manuscript has some problems, and the quality would improve significantly, I think, after revising it significantly (e.g., adding some explanations, removing some logical problems and contradictions, etc.). This might take the authors a few days of extra work, and then the final version would be much better and publishable.

Referee Report on the manuscript:

“Super-fast dynamics and flow instabilities of superconducting vortices”

by L. Embon et al.

In this work, the authors report on the experimental observation of current-driven ultrafast flux dynamics (with velocities up to tens of km/s) in superconducting Pb thin-film-based nanobridges. To detect the flux motion at such high speed, the authors used a SQUID with single-spin sensitivity.

The experimental images and videos are very impressive. The authors were able to resolve the dynamics of vortices at relatively weak driving and their transformation to branching flow patterns at strong driving.

The first overall impression of this work is very positive. The experimental images and videos are quite impressive.

However, a careful reading of the manuscript reveals a number of questions and comments which are listed below.

- 1) One thing that pops-up shortly after starting reading this manuscript is the lack of coherence throughout the text. From the style of the manuscript one can guess that different parts were written by different authors (thought this is in principle fine, of course, taking into account the number of the authors and different affiliations). The problem is that it looks like different contributors did not carefully match the logic used in one part of the text, with the logic used in some other parts of the text. The logical coherence seems to be fragmented (and sometimes contradictory) in several parts.

Some examples (not all are listed here, due to lack of time. There are more inconsistencies in the text):

- The authors formulate a question (on page 2) about the nature of vortices under ultrafast driving:

whether or not these remain well-defined topological defects and if vortex matter at extreme conditions is the same as at low velocities.

This interesting question does not seem to be answered in the text, but then (on page 4) they call the moving flux “flow of quantized vortices”. Yes, this is true for low velocities. But what about ultra-high velocities? Why the results for low (or even medium-range) velocities apply to ultra-high ones? There should be a crossover (or phase transition) when increasing the drive to high enough values. This crossover or transition is not quite clearly described in this work, and it seems to be important to

describe it better, to have a more coherent picture (from the low-drive regime, all the way to very fast drives).

- On page 7, the authors discuss the branching of the flux flow and suggest that this may be related to a “inhomogeneous current density”. Branching, i.e., bifurcations in the flux flow trajectories, generally results from fluctuations; these may either be related to underlying pinning (which is however disregarded in the simulations!) or to fluctuations in the superconducting condensate density. Noticeably, Supplementary Note 5 specifically addresses one of these mechanisms. It discusses pinning-induced branching (which is perhaps the most reasonable mechanism here). However, surprisingly, it looks like this Note is then ignored in the main text on page 7.
- 2) The authors state on page 4 that dendritic-like channels of moving flux reported in this work are different from transient dendritic flux avalanches observed earlier using MO imaging in increasing magnetic fields. Indeed, the driving mechanisms are different in the above two cases. However, does this mean that the observed channels and dendritic avalanches have different morphologies? They look rather similar. A somewhat deeper analysis seems to be needed to draw conclusions about the flux channels.
 - 3) Related to the above point, the authors claim on page 4 that these are vortex chains moving along channels. This statement does not seem to be justified for ultra-high velocities detected in the experiment. The authors themselves note that these velocities greatly exceed the pair-breaking speed limit of the Cooper-pair condensate. At these conditions, no vortices can be formed (as there is no condensate), and one should talk about moving flux rather than vortices. An approach, that takes into account the Joule heating produced by moving vortices, is presented in the Supplementary Note, but ... this is not discussed in the main text, where the authors keep referring to the moving flux to as “vortices”. There are other parts where the logic is inconsistent and/or fragmented.
 - 4) Regarding the simulations.
 - First of all, the TDGL does not seem to be applicable to the flux-motion regime observed in the experiments. Therefore, using TDGL simulations might not be persuasive among readers, even if the simulations reveal any apparent similarity with experiments.
 - Moreover, the reader learns from the text that, although the authors recognize the fact that the TDGL is not applicable (as not only vortices but even the Cooper-pair condensate do not exist at ultra-high velocities), they still use it ... in order “to reveal essential physics” (!). This logic seems to be contradictory. One has to use a suitable tool to explore the ultra-fast flux-flow

regime reported here. The claim in the text, that the TDGL is the best available tool, seems surprising. A proper theoretical tool is required to tackle the problem of flux motion at ultra-high velocities.

- Heating effects and pinning are *key* ingredients in the understanding of the flux propagation along the channels and branching. This idea is accepted by the authors. However, at the same time, heating effects and pinning are *disregarded* in the simulations... "to reveal the essential physics" (!). It is not clear why heating effects and pinning should be disregarded. These effects can be easily implemented, as known from the literature (see, for example, the works of Rakhmanov et al., e.g., Phys. Rev. B 75, 024509 (2007), where both heating effect and pinning are taken into account in the London limit).
- It seems to be difficult to justify the choice of the values of the parameters used: $u=5.79$ and $\gamma=100$.

All of the issues raised in the report, better be addressed in the actual manuscript, which is where it counts, not just in the reply to the referees.

- 5) The list of references misses important related works, especially regarding vortex branching and bifurcations in the flux flow trajectories. These could be easily included, at least in the introduction, because are relevant early references.

Intermittently flowing rivers of quantized magnetic flux, Science 271, 1373-1374 (1996).

C.J. Olson, et al., *Fractal networks, braiding channels, and voltage noise in intermittently flowing rivers of quantized magnetic flux*, C.J. Olson et al., Physical Review Letters 80 (10), 2197 (1998);

C.J. Olson, et al., Superconducting vortex avalanches, voltage bursts, and vortex plastic flow: Effect of the microscopic pinning landscape on the macroscopic properties
Phys. Rev. B 56, 6175 (1997).

A. Mehta, et al., Topological invariants in microscopic transport on rough landscapes: Morphology and Horton analysis of river-like networks of vortices
Phys. Rev. Lett. 82, 3641 (1999).

It could be insightful to compare the experimental results obtained with this technique, with the theory in the above PRLs (for various values of the drive, pinning, etc.) --- although this seems more suitable for future work, but perhaps some quick links could be identified now.

Reference 5 presents one aspect of superconducting circuits, while more general overviews appear in these longer reviews:

- *Atomic physics and quantum optics using superconducting circuits*, J.Q. You et al., Nature 474 (7353), 589-597 (2011);

- *Hybrid quantum circuits: Superconducting circuits interacting with other quantum systems*, Z.L. Xiang et al., Reviews of Modern Physics 85 (2), 623 (2013).

In conclusion, this is a very nice work reporting new observations on the current-driven ultra-fast flux motion in nano-patterned superconducting films that opens new challenges for theorists and further experiments. In my opinion, these findings deserve publication (after all the issues raised above are addressed). A significantly improved manuscript would be very beneficial, because it would have more coherence, avoid some logical contradictions in the text, and avoid some physics problems, some listed above. Therefore, it requires a revision in line with the remarks listed above, before it can be published in Nature Communications. After the above changes are implemented, this improved manuscript would become much stronger, and could then become an important work in this field, and I would then very strongly support its publication in Nature Communications.

Reviewer #3 (Remarks to the Author):

The authors of manuscript NCOMMS-16-25510 present an interesting study of the dynamics of strongly driven vortices in a Pb micro-bridge. The manuscript is well written and the presentation is very attractive. The subject matter of strongly driven vortices and questions about their ultimate stability are interesting to the readership of Nature Communications.

However, there appears to be an inconsistency in the manuscript:

The quantitative analysis of the images in terms of vortex velocity $v(x)$ is based on the relation $v(x) = f a(x)$, where f is the vortex nucleation frequency determined by the voltage across the constriction and $a(x)$ is the vortex spacing. The authors determine $a(x)$ from the average magnetic field $B_{av}(x)$ above the flow channel which is essentially an average over the field B_v of isolated stationary Abrikosov vortices, and obtain vortex velocities of the order of 20 km/s. Such values are very high; for reference, the authors mention that the Larkin-Ovchinnikov instability has characteristic velocities of 1 – 10 km/s.

In the discussion section the authors propose physical scenarios in which vortices could reach these velocity, including slipstreaming in which a trail of depressed order parameter still persists when the next vortex arrives, or strong local heating along the flow channel, which would also locally suppress the order parameter. The authors characterize these scenarios as *self-induced* Josephson junctions. This, however, would imply that vortices moving in the channel are strongly distorted, and that their field profile is not given by that of a stationary Abrikosov vortex (see for instance the shape of Abrikosov-Josephson vortices at grain boundaries). Thus, the use of the profile $B_v(x)$ of a stationary vortex in the determination of $v(x)$ seems inappropriate and the obtained numerical values of the velocity may therefore not be reliable.

Furthermore, one may note that velocities of 20 km/s are indeed extra-ordinary for Abrikosov vortices; for Josephson-like vortices, however, they are much more common. In fact, RSFQ circuits and flux flow oscillators operating at ~ 400 GHz are based on the fact that Josephson vortices can reach very high velocities. Thus, in order to avoid misunderstandings, the authors may wish to mention from the very beginning that they observe at high drive a transition to Josephson-like behavior, and compare their results to the dynamics of Josephson vortices.

The manuscript required extensive revisions prior to publication in Nature Communications.

Response to Referee 1

I have carefully read the manuscript “Super-fast dynamics and flow instabilities of superconducting vortices” by L. Embon et al. (NCOMMS-16-25510) and I found very interesting data with important conclusive results. The manuscript deals with a powerful tool able to visualize at a nanometer scale the motion of quantized vortices driven by electric currents in superconductors. The main focus is the “imaging” of single flowing vortices at high velocities such as tens of km/s, which has not yet been achieved by other direct imaging techniques. The experimental data obtained by such a scanning SQUID-on-tip (SOT) microscopy are accompanied by numerical simulations performed using the generalized TDGL model, that has several limitations and, although qualitatively, it shows the several features of the experimental observations. Apart from the novelty, the claims are convincing and the experiment gives a direct evidence of a channel-like vortex motion into the chosen constricted geometry of the superconducting material under investigation. However, there’s a general comment to be done on this manuscript in order to strengthen the paper further, that is the authors should underline if the obtained results can be generalized, and how? Is the study applicable to any type of superconducting material?, whereas it seems strongly influenced by material parameters.

We thank the referee for the supportive comments. We believe that the results of our work are applicable to vortices driven by strong currents in any type-II superconductors or thin films. Referee made a good point that the SOT technique can also be used for the investigation of intermediate state and the motion of superconductor-normal boundaries in type-I superconductors. This opportunity is now mentioned in the last paragraph of the revised text.

There are several remarks I would like to point out:

1) First of all, concerning the TITLE, it does not sound appropriate since it should include the keyword “Imaging”, for example “Imaging of super-fast dynamics of superconducting vortices”.

We agree with the Referee and have added “Imaging” to the title as suggested.

2) In the ABSTRACT the presented technique is addressed with “single spin sensitivity”, although it has such peculiar characteristic, it has already been used by the authors as a “single vortex” microscopy device, as referred in the proper cited reference [10] of the manuscript.

We agree with the Referee and removed the “single spin sensitivity”.

3) In the Introduction part, the balancing of the driving force and the viscous one can still be done in the presence of a pinning force for $J \gg J_c$? In particular in the presence of a dynamic pinning force which is generally a velocity dependent force, see for example PRL 108, 217001 (2012). Would you comment on the two possible cases of weak and strong pinning?

We agree with the Referee and have modified the text accordingly and added the reference on page 2. We would like to emphasize that the main results of this work are the novel SOT observations of superfast vortices: qualitative theoretical estimates and TDGL simulations are used not for quantitative explanations of the SOT data but only to get insights into this rather unexplored field of vortex dynamics. Generally, we are dealing with the case of strong driving force that is much larger than the pinning force, so it is reasonable to assume that the materials-specific issues of weak and strong pinning become less important allowing investigation of the fundamental mechanisms of nonlinear viscous drag of single vortices.

4) The superconducting material under investigation can be considered a type-II superconductor with a very small Ginzburg-Landau parameter, does this condition influence the observed and obtained results?

On the other hand Pb superconductor can easily be tuned to be type I superconductor, for which it is well known the current-induced resistive state [see e.g. “Magnetic Flux Structures in Superconductors” by R.P.Huebener edited by Springer in Solid-State Sciences], which can have several similarities with the vortex resistive state in type-II superconductors. What would you expect in this case?

This study was performed only on dirty Pb films with the GL parameter around 2. We do not know what would be the effect of the GL parameter on the observed phenomena, but we agree with the Referee that it would be a very interesting topic for future investigations. We have now added this suggestion to the discussion section.

5) In the Introduction part there’s a rather strong claim on the TDGL applicability, but then this is the numerical method that is used in the following sections. The statement “TDGL equations ...are no longer applicable...” should be modified accordingly to Supplementary Note 1.

We agree with the Referee and have rephrased the introduction statement accordingly.

6) In the Introduction part there’s another claim on the possibility to measure the “non constant velocity of vortices” by a local spatial probe, as it is done with this SOT technique, does it mean that the obtained results are in contrast with the results reported in the literature? On the other hand the V-I measurements have been also used by some of the authors in the reference New Journal of Physics 14 (2012) 053006.

Measurements of V-I characteristics is the standard method of extracting vortex velocities, but, of course, it provides only an average velocity. Our work shows that vortex velocities can be highly nonuniform both along the current direction (Fig. 2) and perpendicular to it (Fig. 4b), particularly near the film edge and at low fields. We have now added a paragraph on page 3 that discusses this issue in more detail.

7) In the experiment the field-cooled vortex state is always prepared, have you ever tried zero field cooling as the starting state? Could you show up the results on super-fast dynamics in this case too?

The static vortex configuration is clearly dependent on the history of magnetic field and applied current as seen for example in Movie 1 for various $I < I_c$. But, once vortices are set in motion, particularly by high current drives investigated in our experiments, the magnetic history is “erased”. We did not however perform a systematic study of this effect.

8) In Fig. 3a the V-I curves are shown for voltages up to finite values marked by black arrows. How can these abrupt jumps be ascribed to a thermal runaway and/or not to another mechanism? What is the control of Joule self-heating during such measurements?

We do not have a clear-cut proof that the jump is due to a thermal runaway and have rephrased the statement on page 4 accordingly. Experimentally we observe a jump in the voltage V by more than an order of magnitude and loss of SOT signal. These observations are consistent with a local heating above T_c in a narrow strip across the constriction. A current source was used with no control of the Joule heating. We did not investigate the state above the jump.

9) Moreover in the Discussion other mechanisms are examined, some of them are related to vortex confinement in the channels and others can be responsible of additional buckling of vortex channels itself. However it is not straightforward to follow the Discussion section, which need to be better organized and written, by clearly separating branching instabilities (a secondary effect) from vortex channeling (primary result).

We agree with the Referee and have reorganized the discussion on pages 5 and 6 addressing separately the channeling and the branching phenomena.

10) Among the mechanisms responsible for branching instability there's the Supplementary Note 5 fully dedicated to the pinning influence, in which pinning is treated perturbatively, but in the main section of the manuscript (Discussion) it is stated that the Pb film under investigation revealed dense structure of strong pinning sites as reported elsewhere in their ref. [10]. Is it contradictory in some way?

Strong pinning was implied in the sense of stronger single-vortex pinning by sparse defects relative to the weaker collective pinning by random point disorder. The critical current densities J_c for vortex depinning from these sparse defects is, however, significantly lower than J_d . This can be clearly seen in Movie 1 and Fig. 2f that show that a wide vortex free region is formed already in the stationary situation at applied currents well below I_c . These images demonstrate that vortices escape from the pinning sites well before the local current density reaches J_d . Therefore, our measurements done at local J close to J_d basically probe a nearly free, but highly nonlinear dynamics of vortices as the effect of the pinning defects on superfast vortices becomes weak. In Supplementary Note 5 we provide for completeness an estimate of possible effects of disorder on premature bifurcation of vortex channels. Following the advice of the Referee, we clarified these points on pages 3, 4 and 5. We have also adjusted the discussion on page 6 and revised the Supplementary Note 5.

11) Addressing the vortex structure evolution at superfast velocities, the LO dependence of the damping coefficient is assumed in the calculations, although it is claimed before in the discussion text that the LO instability has not been observed. Is it also contradictory in some way?

We did not postulate the LO dependence of the damping coefficient in our TDGL simulations, but it turned out that the velocity dependence of the nonlinear viscosity of vortices obtained from the TDGL simulations can be described reasonably well by the LO formula. According to our TDGL calculations this occurs at velocities substantially higher than those attained experimentally and hence there is no contradiction. We have not observed an experimental evidence of LO instability under our experimental conditions, but we cannot rule out the LO instability very close to the penetrating edge ($x \lesssim 0.5 \mu\text{m}$) where the velocity is maximal. We have now clarified this point accordingly and restructured the discussion of the LO instabilities on page 7.

12) Concerning Fig.4, which shows the spatial variation of both the vortex velocity and the intervortex spacing for 1 stem, it can be inferred that the velocity become practically uniform within 1/5 of the sample constriction width for this type of geometry, I wonder if this can be generalized to wider strip geometry, in which such a spatial velocity distribution can be neglected?

The extremely high vortex velocities are attained near the penetrating edge of the sample in the region that corresponds to the "vortex-free" region of the geometrical barrier at the conditions of applied current just below the critical current (Refs. 32-35). In this vortex-free region the current density drops from J_d at the edge down to about the bulk J_c value leading to corresponding drop in vortex velocity. In the remaining "vortex dome" region the current density becomes more uniform and so does the vortex velocity. The width of the vortex-free region is geometry, field, and J_c dependent and shrinks upon increasing the field and J_c . The vortex-free region is present also in wider strips although its relative width may be lower. We have now clarified some of these points on page 3.

13) In the Supplementary Fig.1 the six 10 micron wide microbridges with constrictions of different widths are not visible, try to have a better SEM image.

Following the referee's suggestion, we improved the SEM image in the Supplementary Fig. 1.

14) In the TDGL model description included in Methods there is the parameter u which is not defined, and whose value has not been justified, but it contributes to the main parameter τ_{GL} of the simulations.

The value of $u=5.79$ is given by the TDGL theory in the dirty limit (Ref. 9). We have now clarified it the methods section.

15) Based on the actual granularity of the Pb film investigated, authors should comment on the effective influence of weak rather than strong pinning on the branching mechanism of vortex chains.

As pointed out above, at $J \gg J_c$ the effect of granularity and natural pinning defects on the motion of superfast vortices is weak. In turn, the large coherence length in Pb does not suggest any weak link behavior of grain boundaries. For these reasons we believe that our results probe the viscosity-limited dynamics of superfast vortices irrespective to materials-specific features like pinning and granularity. We have now clarified this point on pages 3 and 5.

Finally I suggest that this manuscript deserves publication in Nature Communication, once all the criticism has been satisfactory clarified.

We thank the Referee for his support and very constructive comments.

Response to Referee 2

In this work, the authors report on the experimental observation of current driven ultrafast flux dynamics (with velocities up to tens of km/s) in superconducting Pb thin-film-based nanobridges. To detect the flux motion at such high speed, the authors used a SQUID with single-spin sensitivity. The experimental images and videos are very impressive. The authors were able to resolve the dynamics of vortices at relatively weak driving and their transformation to branching flow patterns at strong driving. The first overall impression of this work is very positive. The experimental images and videos are quite impressive. However, a careful reading of the manuscript reveals a number of questions and comments which are listed below.

We thank the reviewer for the positive assessment of our work.

1) One thing that pops-up shortly after starting reading this manuscript is the lack of coherence throughout the text. From the style of the manuscript one can guess that different parts were written by different authors (thought this is in principle fine, of course, taking into account the number of the authors and different affiliations). The problem is that it looks like different contributors did not carefully match the logic used in one part of the text, with the logic used in some other parts of the text. The logical coherence seems to be fragmented (and sometimes contradictory) in several parts. Some examples (not all are listed here, due to lack of time. There are more inconsistencies in the text):

- The authors formulate a question (on page 2) about the nature of vortices under ultrafast driving: *whether or not these remain well-defined topological defects and if vortex matter at extreme conditions is the same as at low velocities*. This interesting question does not seem to be answered in the text, but then (on page 4) they call the moving flux "flow of quantized vortices". Yes, this is true for low velocities. But what about ultra-high velocities? Why the results for low (or even medium-range) velocities apply to ultra-

high ones? There should be a crossover (or phase transition) when increasing the drive to high enough values. This crossover or transition is not quite clearly described in this work, and it seems to be important to describe it better, to have a more coherent picture (from the low-drive regime, all the way to very fast drives).

The totality of our experimental results shows that vortices remain well-defined topological defects over the entire range of accessible velocities of up to 10-20 km/s. Our TDGL calculations are consistent with this experimental conclusion and indicate that the crossover from Abrikosov to Josephson vortices likely occurs at velocities higher than those currently accessible experimentally (Fig. 6). We have now expanded the discussion and clarified these points on page 7 and in the abstract.

- On page 7, the authors discuss the branching of the flux flow and suggest that this may be related to a “inhomogeneous current density”. Branching, i.e., bifurcations in the flux flow trajectories, generally results from fluctuations; these may either be related to underlying pinning (which is however disregarded in the simulations!) or to fluctuations in the superconducting condensate density. Noticeably, Supplementary Note 5 specifically addresses one of these mechanisms. It discusses pinning-induced branching (which is perhaps the most reasonable mechanism here). However, surprisingly, it looks like this Note is then ignored in the main text on page 7.

As described in detail in Supplementary Note 4, the main mechanism for the observed branching under the experimental conditions of nonuniform vortex velocity along the channels comes from buckling instability as the distance of the slowing down vortices drops below a critical spacing a_c at which the repulsive force between vortices overcomes the confining potential of the channel. This bifurcation mechanism requires neither pinning nor fluctuations, as is indeed confirmed by our TDGL simulations. Supplementary Note 5 points out that pinning disorder can further enhance this buckling instability. We have now clarified this point on page 8 and in the Supplementary Note 5 accordingly.

2) The authors state on page 4 that dendritic-like channels of moving flux reported in this work are different from transient dendritic flux avalanches observed earlier using MO imaging in increasing magnetic fields. Indeed, the driving mechanisms are different in the above two cases. However, does this mean that the observed channels and dendritic avalanches have different morphologies? They look rather similar. A somewhat deeper analysis seems to be needed to draw conclusions about the flux channels.

Even though the vortex channel patterns may look similar, the underlying mechanisms of these two phenomena are fundamentally different. While in our case individual quantized vortices move along stationary paths in superconducting material, dendritic avalanches are basically local thermomagnetic flux jumps driven by strong inductive overheating produced by electromagnetic stray field propagating above the film. The mechanisms of thermomagnetic flux jumps are purely macroscopic and not directly related to the motion of individual vortices. We have now rephrased the paragraph on page 4 to make this essential distinction clearer.

3) Related to the above point, the authors claim on page 4 that these are vortex chains moving along channels. This statement does not seem to be justified for ultra-high velocities detected in the experiment. The authors themselves note that these velocities *greatly exceed* the pairbreaking speed limit of the Cooper-pair condensate. At these conditions, no vortices can be formed (as there is no condensate), and one should talk about moving flux rather than vortices. An approach, that takes into account the Joule heating produced by moving vortices, is presented in the Supplementary Note, but ... this is not discussed in the main text, where the authors keep referring to the moving flux to as “vortices”. There are other parts where the logic is inconsistent and/or fragmented.

This is an essential point that indeed appears counterintuitive at first sight. However, unlike the condensate, which cannot move faster than the pairbreaking speed limit, the velocity of vortices moving perpendicular to the current superflow can be much higher than the pairbreaking velocity, as demonstrated by our experiment and confirmed by the TDGL calculations. This does not contradict any basic principles, similar to the situation with a sailboat that can move much faster than the wind if the boat moves almost perpendicular to the wind direction. We have now clarified this point on page 2 and in the abstract.

4) Regarding the simulations.

- First of all, the TDGL does not seem to be applicable to the flux motion regime observed in the experiments. Therefore, using TDGL simulations might not be persuasive among readers, even if the simulations reveal any apparent similarity with experiments. - Moreover, the reader learns from the text that, although the authors recognize the fact that the TDGL is not applicable (as not only vortices but even the Cooper-pair condensate do not exist at ultra-high velocities), they still use it ... in order “to reveal essential physics” (!). This logic seems to be contradictory. One has to use a suitable tool to explore the ultra-fast flux-flow regime reported here. The claim in the text, that the TDGL is the best available tool, seems surprising. A proper theoretical tool is required to tackle the problem of flux motion at ultra-high velocities.

It seems that this point of Referee is related to the same confusion mentioned in point 3 discussed above. While the condensate can only move with velocity smaller than the pair braking speed ($J < J_d$), quantized vortices can indeed move at much higher velocities without destroying the global superconducting state. This point is now clarified on page 2.

We are well aware of the limitations of the TDGL theory, as was clearly stated in the text. Yet currently TDGL is the only practically available theoretical tool which enables one to calculate self-consistently the dynamics of vortices and distribution of currents and superfluid densities in the mixed state where pairbreaking effects are important. For these reasons, the GL and TDGL theories have been widely used in the literature outside their formal applicability limits to address qualitatively the essential physics of vortices. We have used TDGL only to get qualitative insights into the fascinating experimental observations but by no means for quantitative explanations of the SOT data. We have now clarified these points on pages 2, 6, and 10.

- Heating effects and pinning are key ingredients in the understanding of the flux propagation along the channels and branching. This idea is accepted by the authors. However, at the same time, heating effects and pinning are *disregarded* in the simulations... “to reveal the essential physics” (!). It is not clear why heating effects and pinning should be disregarded. These effects can be easily implemented, as known from the literature (see, for example, the works of Rakhmanov et al., e.g., Phys. Rev. B 75, 024509 (2007), where both heating effect and pinning are taken into account in the London limit).

Based on the totality of our experimental data, we concluded that overheating and flux pinning which can result in the well-known thermomagnetic avalanches and dendritic instabilities are not the key effects behind our SOT results. In our steady state flow regime at $J \gg J_c$ pinning affects very weakly the nonlinear viscosity-dominated dynamics of superfast vortices. Our additional TDGL simulations that included heating and pinning have produced very similar qualitative results so we do not show them here. The purpose of the TDGL simulations in this primarily experimental paper was to present a minimal model that provides qualitative insights into the viscosity-dominated physics of superfast vortices, rather than to carry out a full scale multi-parameter numerical study, which is outside the scope of this paper. We have now clarified this point on page 6.

- It seems to be difficult to justify the choice of the values of the parameters used: $u=5.79$ and $\gamma=100$.

The value $u=5.79$ is given by the TDGL theory in the dirty limit (Ref. 9), while the value of the materials-dependent parameter $\gamma=100$ is an order of magnitude estimate for Pb. In real materials γ depends on many uncertain parameters which determine the broadening of the gap peaks in the density of states unrelated to the inelastic electron-phonon scattering. We have now clarified this on pages 9 and 10.

All of the issues raised in the report, better be addressed in the actual manuscript, which is where it counts, not just in the reply to the referees.

We have incorporated all the relevant clarifications in the manuscript, as mentioned above.

5) The list of references misses important related works, especially regarding vortex branching and bifurcations in the flux flow trajectories. These could be easily included, at least in the introduction, because are relevant early references.

Intermittently flowing rivers of quantized magnetic flux, Science 271, 1373-1374 (1996).

C.J. Olson, et al., *Fractal networks, braiding channels, and voltage noise in intermittently flowing rivers of quantized magnetic flux*, C.J. Olson et al., Physical Review Letters 80 (10), 2197 (1998);

C.J. Olson, et al., *Superconducting vortex avalanches, voltage bursts, and vortex plastic flow: Effect of the microscopic pinning landscape on the macroscopic properties* Phys. Rev. B 56, 6175 (1997).

A. Mehta, et al., *Topological invariants in microscopic transport on rough landscapes: Morphology and Horton analysis of river-like networks of vortices* Phys. Rev. Lett. 82, 3641 (1999).

It could be insightful to compare the experimental results obtained with this technique, with the theory in the above PRLs (for various values of the drive, pinning, etc.) --- although this seems more suitable for future work, but perhaps some quick links could be identified now. Reference 5 presents one aspect of superconducting circuits, while more general overviews appear in these longer reviews:

Atomic physics and quantum optics using superconducting circuits, J.Q. You et al., Nature 474 (7353), 589-597 (2011);

Hybrid quantum circuits: Superconducting circuits interacting with other quantum systems, Z.L. Xiang et al., Reviews of Modern Physics 85 (2), 623 (2013).

The referee made good suggestions for future work which will address a different regime of vortex flow branching and bifurcation near the depinning transition dominated by disorder. We have clarified the differences between these two regimes on page 5 and included some of the relevant references.

In conclusion, this is a very nice work reporting new observations on the current-driven ultra-fast flux motion in nano-patterned superconducting films that opens new challenges for theorists and further experiments. In my opinion, these findings deserve publication (after all the issues raised above are addressed). A significantly improved manuscript would be very beneficial, because it would have more coherence, avoid some logical contradictions in the text, and avoid some physics problems, some listed above. Therefore, it requires a revision in line with the remarks listed above, before it can be published in Nature Communications. After the above changes are implemented, this improved manuscript would become much stronger, and could then become an important work in this field, and I would then very strongly support its publication in Nature Communications.

We thank the referee for his support and the constructive comments.

Response to Referee 3

The authors of manuscript NCOMMS-16-25510 present an interesting study of the dynamics of strongly driven vortices in a Pb micro-bridge. The manuscript is well written and the presentation is very attractive. The subject matter of strongly driven vortices and questions about their ultimate stability are interesting to the readership of Nature Communications.

We thank the Referee for the supportive statements.

However, there appears to be an inconsistency in the manuscript:

The quantitative analysis of the images in terms of vortex velocity $v(x)$ is based on the relation $v(x) = f a(x)$, where f is the vortex nucleation frequency determined by the voltage across the constriction and $a(x)$ is the vortex spacing. The authors determine $a(x)$ from the average magnetic field $B_{av}(x)$ above the flow channel which is essentially an average over the field B_v of isolated stationary Abrikosov vortices, and obtain vortex velocities of the order of 20 km/s. Such values are very high; for reference, the authors mention that the Larkin-Ovchinnikov instability has characteristic velocities of 1 – 10 km/s. In the discussion section the authors propose physical scenarios in which vortices could reach these velocity, including slipstreaming in which a trail of depressed order parameter still persists when the next vortex arrives, or strong local heating along the flow channel, which would also locally suppress the order parameter. The authors characterize these scenarios as self-induced Josephson junctions. This, however, would imply that vortices moving in the channel are strongly distorted, and that their field profile is not given by that of a stationary Abrikosov vortex (see for instance the shape of Abrikosov-Josephson vortices at grain boundaries). Thus, the use of the profile $B_v(x)$ of a stationary vortex in the determination of $v(x)$ seems inappropriate and the obtained numerical values of the velocity may therefore not be reliable.

We agree with the Referee that our analysis would not be appropriate for strongly elongated Josephson-like vortices. In our experimental situation, however, this is not the case and the analysis is valid for the following reasons:

a) Supplementary Figure 4c presents the measured field profile across a stationary vortex showing FWHM width of 0.5 μm which is much larger than $\xi = 46 \text{ nm}$ and $\lambda = 96 \text{ nm}$. This wide profile is a result of the magnetic size of the vortex given by the Pearl penetration depth $\Lambda = 2\lambda^2/d \cong 245 \text{ nm}$ further broadened by the SOT diameter of 228 nm and the SOT scanning height of about 200 nm. Under these conditions both the static and moving vortices are seen by the SOT as magnetic monopoles with flux ϕ_0 irrespective of the size and shape of the core as long as it is smaller than Λ . Accordingly, our reconstruction procedure is insensitive to core deformations, unless they reach a scale larger than Λ which is not the case in our SOT experiments as now clarified on pages 7 and 8. Our TDGL simulations suggest that the highly distorted Josephson-like vortices should form only at velocities that are substantially higher than those attained experimentally. This has now been clarified on page 8.

b) The derivation of vortex velocity does not rely on the detailed vortex profile $B_v(x)$ but only on the line integral of it. Since the areal integral of $B_v(x, y)$ equals ϕ_0 regardless of the vortex core structure, the line integral is not very sensitive to the core structure of the vortex.

c) In addition to the derivation of the local vortex velocity we perform a completely independent evaluation of the average vortex velocity in the stem which does not rely on $B_v(x)$. It is based only on the measurement of the voltage drop on the bridge and on the average vortex distance determined by the applied field. The obtained average velocity of 15 km/s is consistent with the derived local velocities in Figs. 4b and 4c.

We have clarified this point on page 5 in the revised manuscript and included these arguments in Supplementary Note 3.

Furthermore, one may note that velocities of 20 km/s are indeed extra-ordinary for Abrikosov vortices; for Josephson-like vortices, however, they are much more common. In fact, RSFQ circuits and flux flow oscillators operating at ~ 400 GHz are based on the fact that Josephson vortices can reach very high velocities. Thus, in order to avoid misunderstandings, the authors may wish to mention from the very beginning that they observe at high drive a transition to Josephson-like behavior, and compare their results to the dynamics of Josephson vortices.

Velocities of 20 km/s appear extraordinary, yet one of the conclusions of our work is that remarkably vortices apparently remain Abrikosov-like even at these extreme velocities and that the transition to Josephson-like (J) vortices may occur at even higher velocities. Our arguments are as follows:

a) From the measured vortex velocity and the calculated local current density we derive on top of page 7 viscous drag coefficient $\eta = 2.6 \times 10^{-8} \text{ kgm}^{-1}\text{s}^{-1}$ that is of the order of the Bardeen-Stephen $\eta_0 \simeq \phi_0^2 / 2\pi\xi^2 \rho_n = 10^{-8} \text{ kgm}^{-1}\text{s}^{-1}$. This indicates the lack of a qualitative change in the structure of the vortex core even at velocities of the order of 10 km/s, in contrast to J vortices for which viscosity becomes much lower than η_0 .

b) Our TDGL calculations in Fig. 5 show that at the experimentally observed velocities vortices essentially preserve their nearly static structure and display dynamic bifurcation patterns and $B_z(x, y)$ corrugations consistent with the experimental observations. The transition to J-like vortices occurs at substantially higher velocities as described in Fig. 6.

c) As shown in Fig. 6a, J-like vortices form a channel in which the order parameter is strongly suppressed (the central blue channel) by the overlapping vortex cores. This gives rise to a strong confining potential of the channel and to reduced repulsive interactions between vortices preventing channel bifurcation, so that a continuous “self-induced Josephson junction” is formed across the entire width of the microbridge. However, our SOT experiment showed that the channels always bifurcate multiple times indicating strong repulsions of vortices, consistent with the dynamics of Abrikosov-like vortices shown in Fig. 5.

d) Due to their overlap and strong suppression of the order parameter J-like vortices would form smooth channels (Fig. 6) with nearly constant $B_z(x)$. This is inconsistent with the clear corrugation of B_z that we observe experimentally along the channels (see for example Fig. 2i) due to velocity variations of well separated vortices, which was also reproduced by our simulations in Fig. 5.

We have now clarified these points in the abstract, on pages 7 and 8, and in captions of Figs. 5 and 6, drawing a clear distinction between the experimental situation of extremely fast Abrikosov vortices and the predicted transformation to AJ and J vortices at higher velocities.

The manuscript required extensive revisions prior to publication in Nature Communications.

We have revised the manuscript following the constructive comments of the referees.

REVIEWERS' COMMENTS:

Reviewer #1 (Remarks to the Author):

The authors have carefully implemented the submitted manuscript, thus it seems rather improved and it sounds well organized. Although all remarks have been satisfactorily accounted for, the present manuscript paves the way to several questions arising from the fundamental mechanism responsible for the observed fast dynamics of Abrikosov vortices, as well as the direct influence of material parameters on the observations by SOT technique. The paper is actually suitable for publication, however its importance remains within the specific community working in the field.

Reviewer #2 (Remarks to the Author):

This revised version of the manuscript is much better than the previous one. This version is definitely publishable in Nature Communications.

Reviewer #3 (Remarks to the Author):

In the revision of their manuscript, the authors have clarified the different regimes of vortex velocity, i.e., those realized experimentally and very high velocities treated in simulations. They also have explained the assumptions underlying the procedure used for obtaining the vortex velocity. I now recommend the publication of this manuscript in Nature Communications.